# Surface Morphology Analysis of Metallic Structures Formed on Flexible Textile Composite Substrates

**DOI:** 10.3390/s20072128

**Published:** 2020-04-09

**Authors:** Ewa Korzeniewska, Joanna Sekulska-Nalewajko, Jarosław Gocławski, Radosław Rosik, Artur Szczęsny, Zbigniew Starowicz

**Affiliations:** 1Institute of Electrical Engineering Systems, Faculty of Electrical Engineering, Electronics, Computer and Control Engineering, Lodz University of Technology, 90-924 Łódź, Poland; artur.szczesny@p.lodz.pl; 2Institute of Applied Computer Science, Faculty of Electrical Engineering, Electronics, Computer and Control Engineering, Lodz University of Technology, 90-924 Łódź, Poland; joanna.sekulska-nalewajko@p.lodz.pl (J.S.-N.); jaroslaw.goclawski@p.lodz.pl (J.G.); 3Institute of Machine Tools and Production Engineering, Faculty of Mechanical Engineering, Lodz University of Technology, 90-924 Łódź, Poland; radoslaw.rosik@p.lodz.pl; 4Institute of Metallurgy and Materials Science, Polish Academy of Sciences, 30-059 Kraków, Poland; zbigniew.starowicz@gmail.com

**Keywords:** thin films, roughness, wearable electronics, textronics, physical vacuum deposition (PVD), physical vapor deposition, surface profile measurement, profilometer, optical coherent tomography, OCT

## Abstract

This paper compares methods for measuring selected morphological features on the surface of thin metallic layers applied to flexible textile substrates. The methods were tested on a silver layer with a thickness of several hundred nanometers, which was applied to a textile composite with the trade name Cordura. Measurements were carried out at the micro scale using both optical coherent tomography (OCT) and the traditional contact method of using a profilometer. Measurements at the micro-scale proved the superiority of the OCT method over the contact method. The method of contactless measurement employs a dedicated algorithm for three-dimensional surface image analysis and does not affect the delicate surface structure of the measured layer in any way. Assessment of the surface profile of textile substrates and the thin films created on them, is important when estimating the contact angle, wetting behavior, or mechanical durability of the created metallic structure that can be used as the electrodes or elements of wearable electronics or textronics systems.

## 1. Introduction

Textronics and flexible electronics are rapidly developing fields of science. Common areas of interest in these fields include the creation of metallic layers with optimal electrical properties on flexible substrates. Such structures can be implemented in items requiring flexible electronics or in clothing with textronic elements and sensors as well. Good quality thin layers, acting as passive elements or electrically conductive paths [1], are a key parameter for their usage. Defects in metallic layers, such as irregular and heterogeneous structures or damage affecting their continuity, can cause disturbances in electrical conductivity, uneven temperature distribution, and thus local overheating, which in extreme cases leads to the destruction of the system [2,3]. In addition, the surface topography of such structures on a micro scale strongly affects their adhesive interaction with other surfaces [4,5,6,7]. An example can be adhesion to the skin surface in monitoring human vital functions [6,8,9]. This is also important from the point of view of the mechanical durability of electronics elements applied on textile substrates [10]. 

The sensor market is developing rapidly. According to Zion Market Research, the sensor market will be worth $11.2 billion in 2025 [11]. Many of the sensors are integrated with clothing, and act as elements of wearable electronics systems. When they are built in clothes, they always in place on the body. The way they are positioned ensures convenience and discretion for the user. Due to the small size and weight, wearing comfort is also ensured, and the user does not have to place it. Textronic sensors must also have low power consumption, flexibility, as well as reliable sensing performance. The sensors combined with clothes include the following types: gas [12], pressure [13], strain [14], temperature [15], sitting posture [16], or urea [17] detectors. 

According to the literature, the higher the roughness parameters, the smaller the nominal contact surface between the surfaces, and adhesion to the rough surface is reduced [18,19,20,21]. Analysis of surface roughness enables fabrics to be selected on the basis of the properties of the electrically conductive layers formed on their surfaces. In [3], a comparison was made between fabrics with different surfaces and weave structures, showing that the diversity of fabric surfaces causes increased resistance and power loss in the applied layers.

The original ways of creating textronic structures and flexible electronics involve weaving thin wires with low electrical resistance into a textile product during the manufacturing process. These types of techniques are usually complex, and at the same time they do not allow for diverse product functionalization such as surface techniques, printing, sputtering, chemical vapor deposition, and physical vacuum deposition (PVD). These more modern techniques enable the production of metallic conductive layers on the surface of flexible substrates. In each case, however, the metallic layer has a different structure and surface morphology [22].

Surface metrology is the analysis of surface irregularities—i.e., all deviations of the real surface relative to the nominal surface. The real surface is what is obtained as a result of applying a layer on a specific substrate, while the nominal surface is a geometrically perfect product with a specific shape, most often specified in the technical documentation. Unfortunately, the real surface is very difficult to measure, and the nominal surface is impossible to achieve, which means that the measured surface should be analyzed, i.e., observed using a specific measuring method. The purpose of surface description is thus to characterize its geometrical structure [23]. It provides a set of all surface irregularities and gives comprehensive information about the shape of the unevenness, which may be characterized by very large hills and pits. The surface cross-section can be presented in the form of a curve, which depends primarily on the shape of the unevenness in the perpendicular (normal) direction to the reference surface cross-section. Curve analysis allows the process by which the structure was produced to be assessed, as well as predicting the functional properties of the surfaces obtained. 

Traditionally, determining surface morphology involves examining its structure at individual cross-sections using contact methods. Obtaining surface topography on the entire plane by these methods is therefore time-consuming, as it requires information from many parallel measuring sections collected with contact profilometers. They are easy to use, resistant to industrial conditions, able to measure any selected fragment of the surface, and have low sensitivity to vibrations. Contact methods have both advantages and disadvantages [24], which are listed in Table 1.

Contact measurements provide good presentation of surface morphology, but the time needed to gather data is much longer than that required by the contactless methods [25]. Contactless methods are optical methods, which can themselves be divided into the optical confocal method, interferometry, scatterometry, triangulation, and the structured light method [26,27,28,29]. The confocal method assembles a spatial surface image from planes on which light successively focuses during the measurement process. Interferometry is based on the analysis of a fringe image, which characterizes the surface of the tested material. Scatterometry involves the analysis of light, which is scattered from the surface. Triangulation means angular detection of the light reflected from the surface—this method is used to inspect dimensions, distances, and displacements both at the microscale and macroscale. The structured light method is based on the analysis of pattern defects, which are projected onto the measured surface—this method is used to analyze the dimensions and shapes of surface features at the micro- and macroscale.

Roughness tests using a profilometer relate to a micrometer scale. Confocal microscopy measures unevenness in the sub-micrometer and nanometric scale with a resolution of up to 0.1 µm/px. Therefore, the confocal microscope measurement is not an alternative to profilometer measurement. Optical coherent tomography, due to its resolution, also measures roughness parameters in a micrometric scale. For this reason, it can be an alternative method to contact measurement using a profilometer. In addition, due to the flexibility of the measured substrate, the optical method using optical coherent tomography (OCT) has been select by the authors. The choice of the OCT method is made for economic reasons, when no nanometer resolution of the profile is required. In this application, confocal systems are usually very expensive. Real-time measurement is also possible for OCT, due to flexible positioning of its scanning head over the surface of the material moving at a low speed [30]. Depending on the physical phenomena used during the measurement process, the results may depend on the physical properties of the surface [24,25]. 

OCT is a high-resolution imaging technique, which provides depth profiles of inhomogeneous and turbid materials in a contactless and non-destructive manner [31]. The generation of OCT images is based on the measure of the magnitude and the time delay of light reflected back from an investigated sample via an interferometric approach. Within the sample, the light is back reflected at scattering particles, and interfaces between materials with different refractive indices. Through the interferometric detection, OCT allows for a measurement of optical pathlengths, which are related to the geometric structure of the sample. It offers non-invasive optical imaging of structures located in a sample with excellent spatial resolution (< 10 µm). 

OCT is suitable for testing both biological structures and artificial materials, but in the initial period of development of this method, the focus was almost exclusively on organic systems research.

For over 15 years, the OCT has increasingly been used for the study of the structures of materials. It tenders an approach to quantify both surface and internal (overall and local) properties of paper [32], silicon integrated-circuits [33], fiber composites [34], dental composites [35], various kinds of coating, e.g., in pharmaceuticals [36], and electrical components [37]. 

OCT is used not only to study deep structures, but, due to the ability to carry out accurate three-dimensional mapping the topology of the samples, can be used to carry out a visual quality inspection of the surface layer or study surface phenomena, such as the wettability of materials [38,39].

The first study demonstrating the application of OCT in printed functional materials and printed electronics was published by Czajkowski et al. in 2010 [40]. The authors applied ultra-high-resolution time domain OCT (UHR TD-OCT) to evaluate the internal structure of epoxy embedded RF-antenna. The same group later published their work on the use of UHR TD-OCT to study the quality of protective films used in printed electronics [41]. Thrane et al., in 2012 [42], demonstrated the use of TD-OCT in imaging the multilayer structure and identifying defects of Roll-to-Roll (R2R) coatings in polymer solar cells.

Modern OCT tools offer speed-enabling online monitoring of printed devices. Alarousu et al. 2013 [30] used spectral domain OCT (SD-OCT) to monitor the structural surface properties of a moving sample of silver-based electrodes printed on a flexible PET plastic substrate. This reveals the advantage of non-invasive OCT inspection over traditional surface testing methods, especially contact profilometers, which have limitations in the respect of real time working. The pros and cons of the OCT method are listed in Table 1.

The purpose of this article is to evaluate the surface profile of a metallic layer applied to a flexible textile composite through which light cannot pass. The optical method proposed in the paper uses optical coherent tomography, and the authors proved that it could offer an alternative to profilometric contact measurements. 

## 2. Materials and Methods

The parameters used for quantitative assessment of the analyzed surface profile are as follows [43,44] (Figure 1):$R_{a}$—arithmetic mean roughness deviation. This is the value of the arithmetic mean deviation of the residual surface roughness (absolute ordinate values of *Z*(*x*)) within the measuring surface inside the elementary segment *lr*.
(1)$$\;R_{a}=\frac{1}{lr}\int_{0}^{lr}\left|Z\left(x\right)\right|dx\approx \frac{1}{n}\overset{n}{\underset{i=1}{\sum }}\left|Z_{i}\right|$$$R_{p}$—the maximum height of the surface. This is the distance between the highest point and the mean plane inside the elementary segment.$R_{v}$—maximum depth of surface depression, defined as the distance between the lowest point and the mean plane inside the elementary segment.$R_{max}=R_{p}+R_{v}$—distance between the maximum height and the minimum depression of the surface.$R_{q}$—mean square deviation of surface roughness. This is the square root value of the surface roughness deviation within the sampling area within the elemental segment.
(2)$$\;R_{q}=\sqrt{\frac{1}{lr}\int_{0}^{lr}Z^{2}\left(x\right)dx}\approx \sqrt{\frac{1}{n}\overset{n}{\underset{i=1}{\sum }}Z_{i}^{2}}$$

Most of the parameters and functions used in three-dimensional analysis are marked with the letter “S,” as the equivalent of the letter “R” in the profile parameters. Hence, the parameters $S_{a},\;\;S_{p},\;\;S_{v}$, and $S_{q}$ are determined for surface area A as follows:$S_{a}$—arithmetic average surface height $S_{a}$, i.e., the arithmetic average surface deviation from the average surface, being the arithmetic mean of the absolute deviations of the surface height from the average surface. This parameter is defined by the relationship:(3)$$\;S_{a}=\frac{1}{A}\;\iint_{A}\left|Z\;\left(x,y\right)\right|dxdy$$$S_{p}$—arithmetic average of the maximum local surface heights in the area.$S_{v}$—arithmetic average of the maximum local depths of the surface of the surface, defined as the distance between the lowest points and the mean plane in the studied area.$S_{q}$—mean square height, i.e., the mean square deviation of the surface, defined analogously to $R_{q}$ as the standard deviation of the height of the surface irregularity. It is determined from the reference surface, using the formula: (4)$$\;S_{q}=\sqrt{\frac{1}{A}\;\iint_{A}Z^{2}\;\left(x,y\right)dxdy}$$

This paper presents the results of measurements of $S_{a}$, $S_{p}$, $S_{v}$, and $S_{q}$ parameters as the most representative for the description of surface morphology.

### 2.1. Samples

Cordura (Green Site Ltd., Poland) with a surface mass of 250 g/m^2^ was used as both the raw material and the substrate for a thin silver layer. This composite textile material consists of nylon threats which are laminated with a polyurethane layer [45]. 

A thin metallic layer was created in the PVD process by thermal evaporation in the Classic 250 chamber of a Pfeiffer Vaccum system. The following parameters were used in the PVD process:Initial vacuum—5 × 10^−5^ mbar;Time of metal deposition—5 min;Deposited metal—Ag with 99.99% purity (guaranteed by Mint of Poland Ltd.) (boiling point 2162 °C), evaporated from the tungsten boat (melting point 3410 °C);Distance between the source of silver particles and the substrate—6 cm;Time of initial conditioning—2 h;Humidity of conditioning—55%;Temperature of conditioning—22 °C.

Thermal evaporation is one of the cheapest PVD technologies that takes place in a relatively short time. On the other hand, it is not possible to deposit materials that boil at high temperatures, such as titanium (boiling point 3287 °C) or chromium (2572 °C). Therefore, the authors of the work did not use these materials to increase the adhesion of the silver coating to the substrate.

The thickness of the deposited metallic layer was measured indirectly. In the vacuum deposition process, the laboratory glass was placed next to the tested substrate in the same technology process. The metal was deposited on both surfaces. Then, using the contact profilometer, the thickness of the layer on the glass (treated as the reference one) was assessed as the height of the measuring needle step. The thickness of the silver applied layer was estimated as 250 nm.

The samples were measured using two distinct techniques: contact and contactless.

### 2.2. Surface Topography Measurement by the Contact Method

Figure 2 shows the principle for creating a 2D profile of the tested surface. A measuring arm tipped with a diamond needle, with a rounding radius of 2 µm, is pressed with a force of 1 mN as it slides over the measured surface. Changes in the position of the needle on the vertical axis, caused by the structure of the surface, are converted into an electrical signal, then amplified, filtered, and processed. If the pressure force of the needle is inappropriate for the hardness of the material, then furrows are formed on the elements of the structure with too low hardness [24].

Gaussian filters are the most often used filter in the study of the geometric structure of surfaces. This type of filter is defined using cut-off wavelengths and calculated on the basis of the Fourier transform. The selected cut-off value determines which components of the original profile or surface will be moved and presented as roughness, and which will be blocked. Our method is subject to normalization and associated with roughness parameters such as $R_{a}$ and $R_{z}$ (for random profiles and surfaces, or for periodic profiles and surfaces—a parameter determining the average value from roughness intervals occurring in the elementary segment interval) [46,47,48]. For surface topography measurements, a Hommel model TurboWaveline 60 model is used (Hommel Hercules Werkzeughandel GmbH, Germany), on which a probe with a measuring tip with a rounding of 2 µm is mounted. This is a laboratory class device. Data from the measuring instrument are processed using software provided by the manufacturer and presented in the form of plots and maps of surface topography. The measuring stand is presented in Figure 3.

The profilometer with Turbo Wave software allows for measurement of roughness and waviness. The measuring device has an automatic moving table (D), which makes it possible to make several measurements along the sample. Using special Hommel Map software, on the basis of several passes of the measuring needle, the computer creates a spatial topographic map of waviness (i.e., irregularities of a random nature or close to the periodic form, the intervals between which significantly exceed the surface roughness intervals), and using the Gaussian filter surface roughness profile (the set inequalities arising as a result of machining, characterized by a small distance between the vertices of height R), it creates a spatial topographic map of the tested object (Figure 4) [49]. 

Topography measurement of the surface of the sample top layer was carried out for 41 lines on a 5 × 5 mm surface with intervals of 125 µm.

### 2.3. Topography Measurement Using the OCT Method

The infrared tomographic system HR Spark OCT 800 nm (*Wasatch Photonics* Inc., NC, USA) [50] presented in Figure 5 was used to capture the volumetric data of Cordura samples. 

The OCT setup contains a Fourier domain (FD) Michelson’s interferometer. In this device, the signal of light interference between the sample and the reference arms at different depths below the fabric surface corresponds to different frequencies of the scanning source spectrum, distinguished by the internal diffraction grid. The sample arm of the interferometer was equipped with both a scanning MEMs mirror and a CMOS camera, which provided fundus images of a material surface. A Computational Engine (PC) containing an image acquisition card was connected to the OCT Engine via a Camera Link interface. The card registered 3D images of the infrared signal, reflected from the material surface or in the semi-transparent subsurface layers. The HR-Spark apparatus creates an array of 1024×1024 A-scans with a resolution of ≈5.2 μm and with scan rates of 76,000 lines/s. The 3D data is obtained from a 2.3 mm scanning area, as a set of 1024×1024×512 voxels with an axial resolution of 1.79 μm (Table 2). The scanning depth of the translucent layers can be up to 1 mm. The images were saved as DICOM files, with the file headers including scale factors in both horizontal and vertical directions, expressed in mm/pixel units. This enabled the evaluation of physical sizes. Four images were acquired for each Cordura type (with and without a metallic layer).

The sample of measurement material was glued to the glass substrate and then placed horizontally, under the OCT scanning head. The tilt of the OCT head was adjusted to provide horizontal orientation of the material surface cross-section in each B-scan (*XZ*-plane) inside the registered OCT image. A typical perspective view of the scanned material covered with the silver layer and a single cross-section (B-scan) are presented in Figure 6a,b, respectively. 

The proposed algorithm for the measurement of surface roughness parameters was developed in the Matlab 2015 environment [51]. It consists of six steps, described below.Reduction of built-in speckle noise by diffusion filtering in the OCT image space.Detection of the surface edge voxels following global image thresholding.Completion of missing edge voxels by spline approximation.Identification of the surface base plane approximated by completed edge voxels.Elimination of outlying edge voxels on the material surface.Approximation of the material surface edge by splines.Evaluation of the basic roughness parameters inside five square windows, adjacent to each other horizontally on the *XY* image plane, each with an area of 1250 × 1250 mm.

Speckle noise, induced by the laser beam scanning the tested material, appears as a strong and frequent variation in image pixel intensity. Such noise is an integral part of every OCT image [52]. The speckles are generated by random interference between mutually coherent waves of infrared light, which are reflected from material inhomogeneities. The same waves also carry information about the material structure. Because speckle noise is multiplicative versus image intensity, the noised intensity image $I^{\prime }\left(x,y,z\right)$ in the original form is expressed as
(5)$$\;\forall p\in D,\;\;I^{\prime }\left(p\right)=I\left(p\right)\left(1+h(p\right))$$
where I denotes a noise-free image, h is white noise with a zero mean and a certain distribution, p=(x,y,z) is an image voxel in the domain D=([0,X−1],[0,Y−1],[0,Z−1]). The firmware of the applied Spark OCT 800 nm system performs a post-scan logarithmic transformation of raw intensity images to reduce their grey-level dynamics. Thus
(6)$$\forall p\in D,\;\;\log(I^{\prime }\left(p\right))=\log\left(I\left(p\right)\right)+\log\left(1+h\;\left(p\right)\right),$$
can be written as
(7)$$\forall p\in D,\;\;I^{\prime }\left(p\right)=I\left(p\right)+h\;\prime \left(p\right),$$
where h′ is white noise of zero mean. To reduce speckle noise and to preserve the original shape of the material surface, median or diffusion filtering can be used [53,54]. We applied an isotropic diffusion filter in the whole image space. This filtering is built into the class itkCurvatureAnisotropicDiffusionImageFilter [55] included in the ITK (Insight Segmentation & Registration Toolkit) online library [54], which has been designed for medical image analysis. It performs the modified curvature diffusion equation (MCDE) [56] and can be directly called in the Matlab environment [51] using the MEX file package Matitk, available on the Internet [57,58]. The filter has the form shown in Equation (8):(8)$$I^{\prime }=matitk\left(\prime FCA^{\prime },\left[iter,step,cond\right],I\right),$$
where $\prime FCA^{\prime }$ denotes the diffusion filter name, iter is the number of iterations in the diffusion algorithm, step<0.0625 represents the time step value of the diffusion process limited by the CFL (Courant-Friedrichs-Levy) convergence condition of the numerical solution of the internal PDE (Partial Differential Equation) [59], and cond is the value of diffusion conductivity. 

The tested Cordura material covered with a silver layer strongly scatters the OCT scanning beam even on the border of the layer. This gives a bright edge to the material surface in each B-scan image, as shown in Figure 6b. Therefore, the next step in the proposed algorithm uses global image thresholding followed by detection of the first edge voxel B(x,y) of the material encountered in the z direction, as presented in Equation (9). The thresholding level T is selected experimentally, once only for the whole class of examined images.
(9)$$B\left(x,y\right)=min_{z}\left(I\left(x,y,z\right)>T\right),$$

Due to the distortions and changes in the material structure that can occur during OCT scanning, the bright material edge may disappear at some locations inside the view field. Thus, the surface boundary values B(x,y) in Equation (9) may be undefined at some (x,y) positions, where the boundary voxel is not found at an acceptable depth. This case is exemplified in Figure 7a. 

To obtain a regular grid of voxels evenly distributed on the material surface, the fast-smoothing spline method described in [60] was selected for data approximation with a minimal smoothing level. This method is based on forward and inverse discrete cosine transform (DCT), and can be expressed as an iterative convergent process defined in Equations (10) and (11).
(10)$$B^{\left(k+1\right)}=DCT^{-1}\left(G^{\;N\;}{}^{\circ}DCT\left(W\;{}^{\circ}\left(B\left(x,y\right)-B^{\left(k\right)}\left(x,y\right)\right)+B^{\left(k\right)}\left(x,y\right)\right)\right),$$
(11)$$G^{\;N}=1÷\left(1^{\;N}+sL^{N}{}^{\circ}L^{N}\right),$$
where N=2 is data dimensionality, DCT, $DCT^{-1}$ denote discrete cosine transform and its inverse, respectively, W is the weight matrix of $w_{ij}\in \left[0,1\right]$ for B(x,y), $G^{\;N}$ is a diagonal matrix of nonzero values, the symbol ° denotes the Schur product, the symbol ÷ is the division element by element, and $L^{N}$ denotes the following *N*-rank tensor: (12)$$L_{i_{1},i_{2},\ldots ,i_{N},}^{N}=\overset{N}{\underset{j=1}{\sum }}\left(-2+2cos\frac{(i_{j}-1)\pi }{n_{j}}\right),$$
where $n_{j}$ is image size in dimension j. In Equation (11), the symbol s represents a positive smoothing factor, which, as it rises, increases the smoothness of the surface approximation. This method is applied in the case of B(x,y) with missing outlier data, by assigning zeros to their weights in the matrix W. The solution of Equation (6) is obtained when the specified approximation tolerance or maximal number of iterations has been reached. The smoothing defined in Equation (12) was performed by implementing the Matlab function given in Equation (13),
(13)z(x,y)=smoothn(B(x,y),W(x,y),s),
where B, W, s have the same meaning as in Equation (10), Equation (12) and the value of s=0.5 corresponds to the minimal level of smoothing. The continuous edge z(x,y) is presented in Figure 7b. Using the surface voxel grid z(x,y)  the surface model can be determined as a plane approximated by least-squares fitting of the data z(x,y) with first order polynomial of two variables x,y. This approximation task uses the Matlab Central functions polyfitn and polyvaln, shown in Equation (14), which are available in [61].
(14)$$p_{11}=polyfitn\left(\left[x,y\right],z,n\right),\;z_{11}\left(x,y\right)=polyvaln\left(p_{11},\left[x,y\right]\right),$$
where n=1 is the polynomial order and [x,y] denote the Matlab style array of image coordinates x=[1,2,…,X], y=[1,2,…,Y], respectively. The example B-scan profiles of the approximated base plane have been illustrated in Figure 8 as straight green lines. 

The difference in height of the surface base plane $z_{11}\left(x,y\right)$ and the edge data z(x,y) define the surface roughness as expressed in Equation (15)
(15)$$z_{R}\left(x,y\right)=z_{11}\left(x,y\right)-z\left(x,y\right).$$

The surface profile outliers of $z_{R}\left(x,y\right)$ were additionally limited to the value ∆z, measured relative to the material base plane $z_{11}\left(x,y\right)$. This was done by repetition of spline smoothing, as expressed in Equation (13) with the different weight array $W_{1}$ satisfying the condition given in Equation (16):(16)$$W_{1}\left(x,y\right)=\{\begin{array}{l} 1\;if\;\left|z_{R}\left(x,y\right)-z_{11}\left(x,y\right)\right|<∆z\;\\ 0\;otherwise \end{array}.$$
then
(17)$$z_{R}^{\prime }\left(x,y\right)=smoothn\left(z_{R}\left(x,y\right),W_{1}\left(x,y\right),s\right),$$
where ∆z=100 μm and B, $W_{1}$, s have the same meaning as in Equation (13). To obtain compatibility with the profilometric measurement, the waviness components of $z_{R}\left(x,y\right)$ were eliminated using lowpass Gaussian filtering, as shown in Equation (18)
(18)$$\;z_{R}^{\prime }\left(x,y\right)=z_{R}\left(x,y\right)-imgaussfilt\left(z_{R}\left(x,y\right),\;\sigma \right),$$
where imgaussfilt(·) denotes the Matlab built in two-dimensional Gaussian filtering of the surface $z_{R}$, and σ is the Gaussian function standard deviation common for vertical and horizontal directions. It has been assumed that σ = 4 s, where s represents the size of any measurement square shown in Figure 9a. This assumption makes it possible to leave the basic surface harmonic unchanged, with a period fitting the measurement square.

Two-dimensional and perspective views of the final depth map $z_{R}^{\prime }\left(x,y\right)$ for an example material surface are shown in Figure 9a,b, respectively.

## 3. Results and Discussion

### 3.1. Control Measurement

In order to assess the accuracy of each presented measurement method, the roughness parameters of reference sample No. 178–601, serial no. 318411402 (Table 3) were analyzed with the profilometric contact method, and the contactless method using OCT. The reference sample is made of alloy stainless steel. Optical coherent tomography was performed using the same method as for the measurement of the Cordura sample, on five selected surface fragments (Figure 10). The components of the sample profile in the OCT image corresponding to waviness were extracted using the Gaussian filter and removed from the surface map, as in the profilometric method. Using the concept of surface profile roughness and waviness components given in [62], it was assumed that the cut-off frequency on the surface image $f_{cut}=4\sigma_{f}$, where $\sigma_{f}$ = 0.25 is the standard deviation of the Gauss filter function against the frequency spectrum from Figure 11. This discrete spectrum refers to a measurement distance of n≈240 pixels, equal to the side length of the measurement square in Figure 9 or Figure 10. The applied waviness filter uses frequency samples spaced four times more densely than in the discrete spectrum computed along a row of any measurement square in Figure 9. In the image space domain, this corresponds to discrete convolutions along a distance four times greater than the row length. At a distance of $4\sigma_{f}$ from the zero-frequency component, the attenuation of the Gaussian waviness filter will be about 70 dB, so the suppression of the basic frequency in the roughness band can be ignored.

The operation of waviness removal in the image domain can be written as:(19)$$\forall_{y},\;z_{R}^{\prime }\left(x\right)=z_{R}\left(x\right)-\left[z_{R}\left(x\right)*G\left(x,\sigma_{x}\right)\right],$$
where * denotes the convolution operation, $G\left(\cdot \right)=F^{-1}\left(G_{f}\left(\cdot \right)\right)$ is the inverted transform of the Gaussian filter $G_{f}\left(\cdot \right)$ creating the surface weave mask $z_{R}\left(x,y\right)$ for the determined line *y*, and $\sigma_{x}$ is the standard deviation of the Gaussian curve along the line associated with the standard deviation $\sigma_{f}$ in the frequency domain using the relationship $\sigma_{x}\cdot \sigma_{f}=n/(2\pi$), where n is the number of measurement samples.

Values for the *R_a_* parameter calculated by the OCT method are averaged for the lines in individual measurement fields covered by the frames shown in Figure 10. The *R_max_* value given in Table 3 was also calculated as the average of the maxima for each image line in five measurement fields. In the case of the reference surface, the roughness values given by the manufacturer and measured by both methods are similar. It should be taken into account that the surface resolution of the OCT method (5.3 μm) is significantly lower than that obtained using the profilometric method (2 μm). The *R_max_* value determined in the contactless method is clearly overestimated compared to the given standard, probably due to impurities and reflections from the shiny metallic surface. Reflective surfaces can cause OCT image distortions over metal surface visible in the air layer in the form of bright vertical streaks. This phenomenon can be partially reduced by shifting the focus range towards the material surface. Then, however, the material border is slightly blurred. Individual, small reflections of IR light can appear over different surfaces and affect *R_max_* value.

### 3.2. Measurement with a Profilometer

Measurements of the Cordura surface profile were made in relation to the reference profile realized by the device rails. After leveling the profile, i.e., separating the mapped slope from the profile, which results from the lack of parallelism of the object surface to the measurement line, the original surface profile was obtained. The measurement results, showing the appearance of the real surface without Gaussian filtration, are presented in Figure 12.

On the surface of the clean composite, pits with high values can be observed. These result in a higher surface roughness value, which is confirmed by the measurement results presented in Table 4. This structure of the substrate can allow a metallic top layer with greater durability to be obtained.

Figure 13 shows the results of measurements in the form of topographic maps with Gauss filtration of the surface roughness of the textile material used as a substrate, with and without a thin metallic layer.

Presented views of the surface roughness and structure of the base material indicate high values for the *S_a_* and *S_v_* parameters. The maximum roughness profile height for the coated layer was 46 µm, while a value of about 95 µm was obtained for the raw surface.

The results presented in Table 4 are mean values of 41 measurement sessions. Analysis of the data shows that a clean substrate is characterized by nearly three times higher values for $S_{a},\;S_{v},\;S_{q}$ parameters in comparison to the substrate with an applied metallic layer. This indicates the greater unevenness of the substrate material itself. The $S_{p}$ parameter is also higher for a clean substrate, although the ratio of the values for a substrate without and with a coating is about 1.5. This parameter describes the arithmetic mean of the local hills on the surface. It therefore results from the fact that the metal is deposited mainly in the local pits of the structure.

### 3.3. OCT Results 

Based on the surface profile $z_{R}\left(x,y\right)$, selected features of material surface roughness were evaluated according to the ISO 25178 standard [63] in five non-overlapping squares with the dimensions 1250×1250 μm centered in the *XY* plane of each OCT image, as illustrated in Figure 9a. The values obtained for the considered area roughness parameters ($S_{a},S_{p},S_{v},S_{q}$) measured from the OCT images are presented in Table 5.

### 3.4. Result Comparison

The results of roughness measurements obtained with both a profilometer and optical coherent tomography relate to measurements on a micrometric scale. Figure 14 summarizes four surface roughness indicators measured with each method.

We observed the differences between the values obtained using the contact and contactless methods. These differences related both to the profile of the clean textile composite substrate and to the profile of thin metallic layer applied to the substrate. The values obtained using the optical sensor (OCT) were higher than those obtained using the stylus method, which has also been reported as being the case for surface structures of composites with an aluminum layer tested using contact and non-contact (optical) methods [16]. For raw substrate, the parameters determined by the optical method were $2\times (S_{p})$, $2.7\times (S_{q})$ and $2.9\times \left(S_{a}\right)$, as high as in the contact method. However, the value $S_{v}$, referring to the distance of the lowest point from the average plane in the sampling area of raw surfaces, was very similar. The reasons for these discrepancies are related to the behavior of the flexible material, Cordura, when it is in contact with the sensor. Since there is a mechanical contact between the surface and stylus, it can deflect the material or even cause damage to the surface polyurethane layer. 

The roughness parameters of the metallic layer measured by the non-contact method were also higher ($1.7\times S_{p}$, $2.3\times S_{v}$, $4.5\times S_{q}$ and $5.4\times S_{a}$) than those measured by the contact method. The reason for such significant differences is the scraping of the metal layer by the measuring stylus in the contact method. An image of a metallic structure measured using a contact profilometer is shown in Figure 15. Scratches in the metal layer can be observed in the form of parallel lines, resulting from the pressure of the measuring needle. Images taken by OCT of the cross-section of the material also show regular metal loss where the profilometer needle passed. Areas with increased brightness correspond to losses in the metallic layer, caused by the action of the profilometer needle (Figure 15b).

The values $S_{p},S_{v}$, which are unique in every measurement square, are not very reliable in the OCT method, because of the image speckle noise reduced by diffusion filtering, and due to approximations applied to the surface profile with discontinuities. The integral-type parameters such as $S_{a},S_{q}\left(R_{a},R_{q}\right)\;$ can be determined with greater accuracy. Another reason for the differences in roughness parameters shown in Figure 14 may be the different diameter of the measuring needle ≈2 μm and the OCT laser beam width ≈5.3 μm, corresponding to the lateral resolution presented in Table 2. Additionally, the precision of surface roughness measurement is limited by the OCT axial resolution of 1.79 μm/pixel.

Using the proposed optical method of assessing the surface profile, discontinuities and coming through defects of the metallic layer can be also observed, if they are at least the size of the device lateral resolution equal to 5.4 µm. They occur in locations where you can see lighter vertical lines of the echo created when IR beam penetrates under the material surface (Figure 15b). The IR light cannot penetrate the continuous metal layer, so in the image in Figure 15c, the area below the surface border stays black. 

## 4. Conclusions

In this study, the authors compared two methods for measuring parameters characterizing the surface morphology of thin metallic layers formed on a flexible textile substrate by PVD: a typical contact method using a profilometer and a contactless method using an optical sensor (OCT). The experiments confirm the usefulness of the OCT method for assessing surface parameters of layers applied on flexible substrates, as an alternative to contact measurements. 

In contact profilometry, the stylus profiler mechanically touches the surface, which can cause damage to the surface of the flexible sample when the contact pressure cannot be precisely adjusted to the hardness of a deposited layer on a composite material. On the other hand, OCT can be regarded as an approximate method because of its own limitations. OCT extreme parameters often are overestimated and have poor repeatability, due to the influence of unwanted infrared light reflections from the surface during measurements. The lateral resolution of High Resolution (HR) Spark OCT device used in tests is about two times worse than in the contact method, which also affects the measurement results. However, the OCT method does not influence the tested metal layer, and thus delivers results unchanged by mechanical factors. Furthermore, it provides higher acquisition speed compared to mechanical scanning of the sample area. 

The studies have been conducted on samples without cracks or delaminations to assess the profile of a uniform surface. Nevertheless, the measurement algorithm can take into account the location of defects in the case of their occurrence. Using this method, it is possible to assess the quality of the created layers by measuring the number and size of discontinuities, but only these extending from the metal surface to the textile substrate. This problem will be studied in the future.

The OCT method can be applied not only for conductive layers, but also for other surfaces, with particular emphasis on flexible materials. Due to the ability of the IR light beam to penetrate semi-translucent or turbid layers, the OCT method can be applied to analyze the profiles of internal surfaces covered by thin polymer or textile layers. This feature enables the inline quality validation of textronic sensors in the industry production conditions.

## Figures and Tables

**Figure 1 sensors-20-02128-f001:**
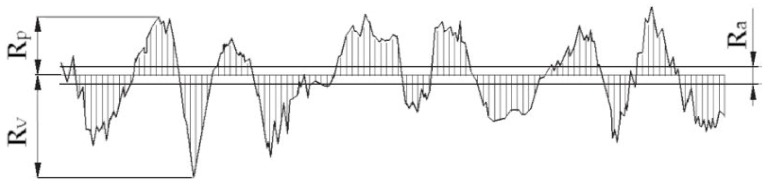
Graphical interpretation of *R_a_*, *R_v_*, *R_p_* parameters inside the elementary segment.

**Figure 2 sensors-20-02128-f002:**
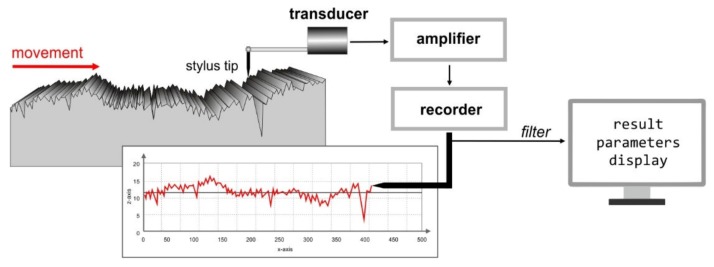
Principle of surface roughness measurement using the contact method.

**Figure 3 sensors-20-02128-f003:**
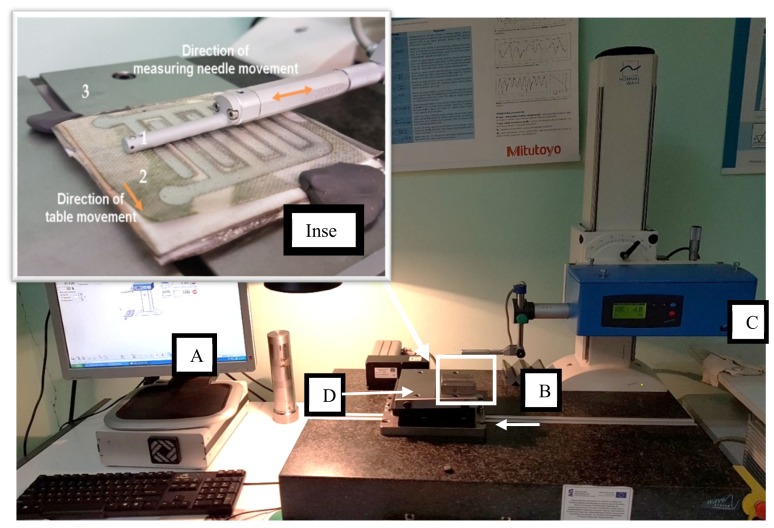
Measuring stand for contact surface topography measurement: **A**—computer with software; **B**—measured sample; **C**—TurboWaveline60 profilograph; **D**—sliding table. Inset: method and directions of measuring the topography of the surface layer of the tested material: **1**—measuring needle; **2**—sample; **3**—moving table.

**Figure 4 sensors-20-02128-f004:**
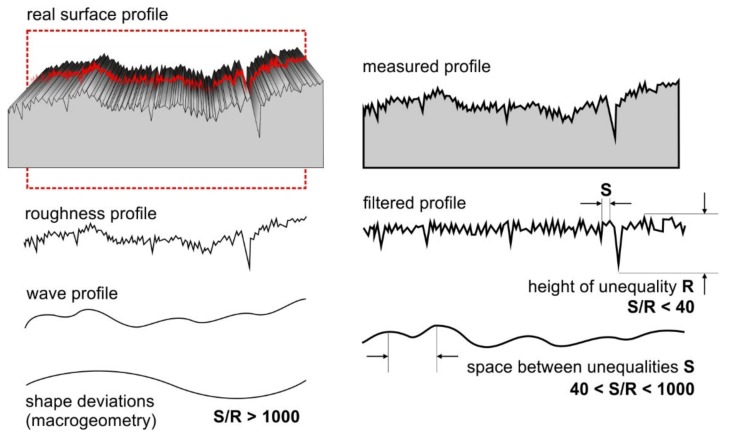
Components of geometric surface structure. S—space between unequalities, R—hight of unequality.

**Figure 5 sensors-20-02128-f005:**
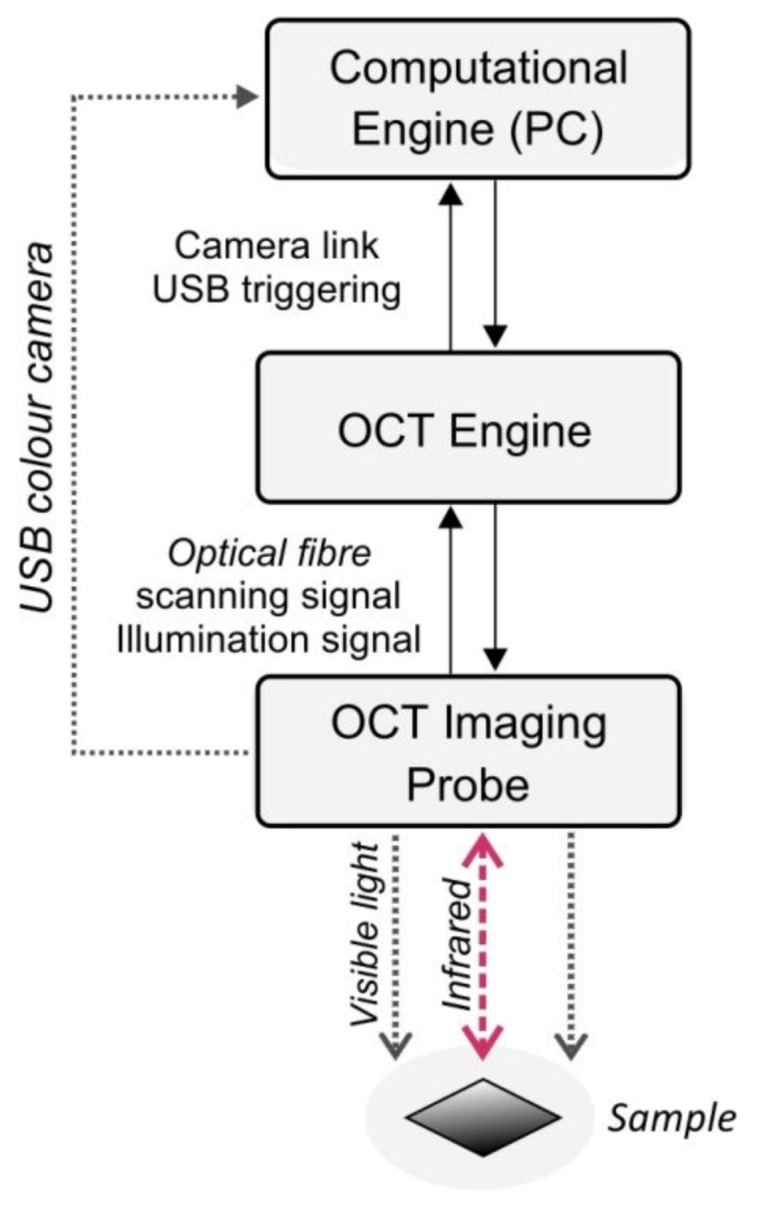
Optical coherent tomography (OCT) setup.

**Figure 6 sensors-20-02128-f006:**
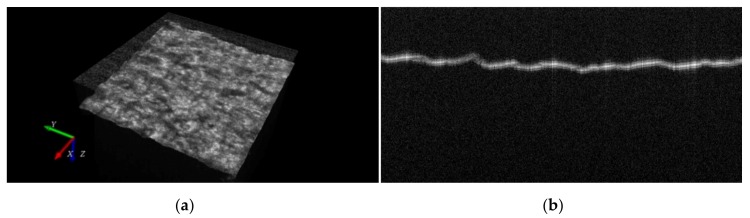
Examples of Cordura images with a silver layer scanned by OCT: (**a**) a perspective view; (**b**) a cross-section (B-scan).

**Figure 7 sensors-20-02128-f007:**
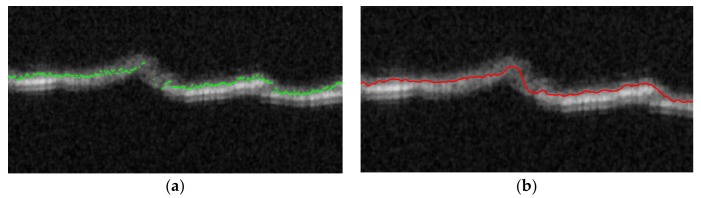
Enlarged fragment of example B-scan of Cordura with an Ag layer: (**a**) with the detected set of edge voxels; (**b**) with a continuous edge approximated with splines.

**Figure 8 sensors-20-02128-f008:**
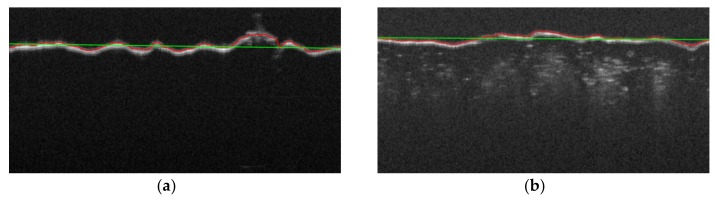
Example B-scan with approximated profile of base plane: (**a**) for Cordura with an Ag layer; (**b**) for Cordura without a metal layer.

**Figure 9 sensors-20-02128-f009:**
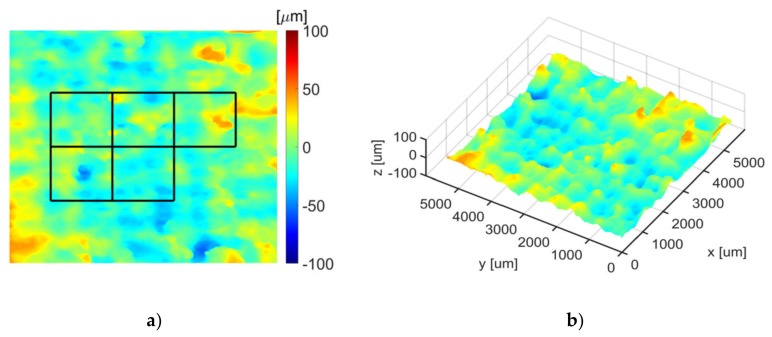
Example depth map of Cordura surface with an *Ag* layer, scanned in the OCT view field relative to the approximated base plane: (**a**) plane view; (**b**) perspective view.

**Figure 10 sensors-20-02128-f010:**
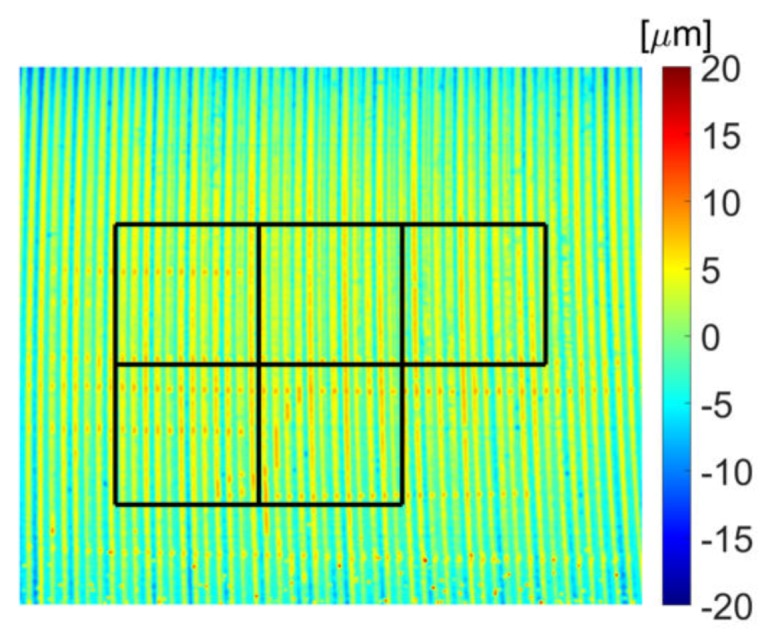
Example map of roughness template No. 178–601 with marked squares showing where the measurement was made.

**Figure 11 sensors-20-02128-f011:**
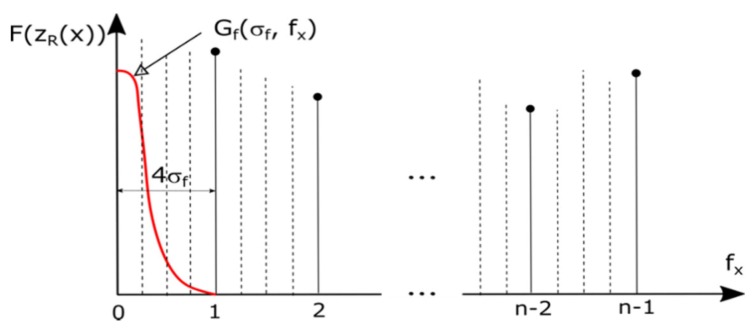
Explanation of waviness elimination by Gaussian filtering in the OCT surface map frequency domain along the horizontal direction: $G_{f}$—waviness selecting Gaussian filter profile; $\sigma_{f}$ —Gaussian filter standard deviation in the frequency domain; *n*—number of roughness map samples inside any measurement square in Figure 9a or Figure 10; $F\left(z_{R}\left(x\right)\right)$ —Fourier spectrum magnitude for the fixed row *y* inside the surface map $z_{R}\left(x,y\right)$.

**Figure 12 sensors-20-02128-f012:**
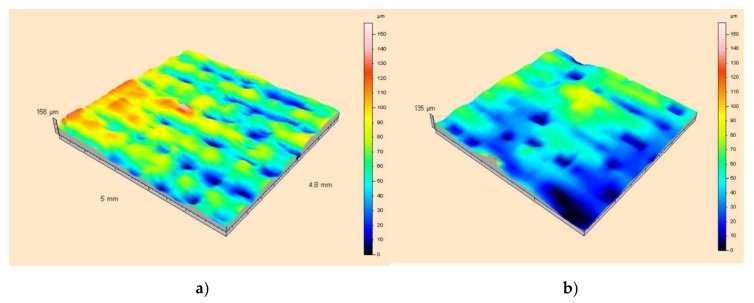
Primary profile (**a**) for the composite surface, (**b**) for the applied coating.

**Figure 13 sensors-20-02128-f013:**
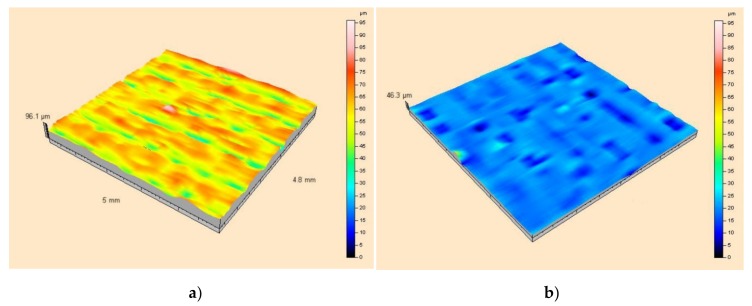
Topographic 3D maps after Gaussian correction, showing surface roughness: (**a**) for a raw surface; (**b**) for a surface with a silver layer.

**Figure 14 sensors-20-02128-f014:**
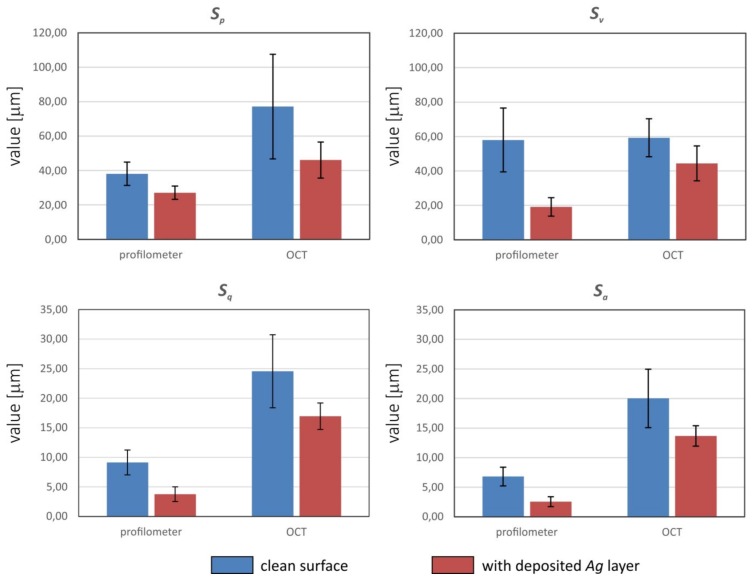
Comparison of the roughness parameter values obtained with two methods: using a profilometer and OCT analysis.

**Figure 15 sensors-20-02128-f015:**
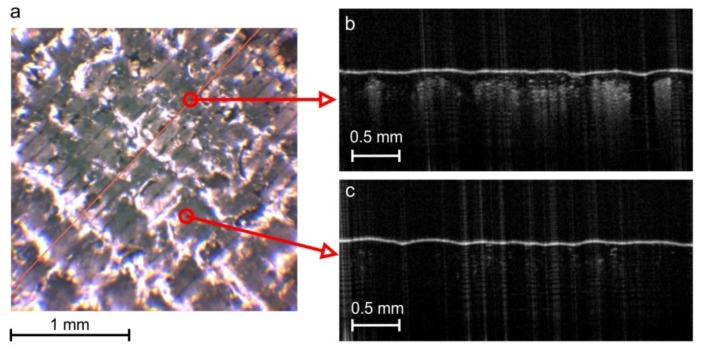
Surface view of the metallic layer formed on a flexible composite substrate after profile measurements using the contact method: (**a**) camera image with visible parallel furrows formed during the measurement; (**b**) OCT cross-section of the place where the profilometer measuring needle passed; (**c**) OCT cross-section between the measuring needle movement paths.

**Table 1 sensors-20-02128-t001:** Comparison of contact method and optical coherence tomography (OCT) method.

	Advantages	Disadvantages
Contact method	Full reflection of the measured surface. No influence of surface optical characteristics. Good penetration of the tested surface. High sampling resolution up to 0.1 µm/px.	The destructive impact of the measuring needle on the tested surface. The needle leaves traces in the materials of low hardness. High level of noise generated by the mechanical system.
		Long measurement time, movement speed of the needle as low as 0.1 mm/s. Possible physicochemical reactions may occur between the needle and the tested material.
OCT optical method	No impact of the measurement on the tested surface.	High influence of surface reflectance characteristic on the measurement results.
	Short measurement time with scanning speed ≈10 mm/s.	Lower sampling resolution of 5.4 µm/px in the case of Spark OCT scanning.
	No traces left on low hardness surfaces. No reaction physical or chemical reaction between low power laser needle (2mW) and the tested surface.	High level of speckle noise in typical OCT images. Possible false peaks and gaps in the surface profile requiring approximation.
	Easy detection of breaks and holes in the metal coating of a textile by the IR light penetrating below the surface. Detection of internal surfaces under semi-translucent coatings.	

**Table 2 sensors-20-02128-t002:** Characteristics of Cordura OCT images.

Characteristic		Cordura without Modification img1–img4	Cordura with Ag Layer img5–img8
Image size (x × y × z)	[px × px × px]	1024 × 1024 × 512	
Lateral resolution (x axis)	[µm/px]	5.30	5.14–5.30
Lateral resolution (y axis)	[µm/px]	5.20–5.22	5.03–5.30
Axial resolution (z axis)	[µm/px]	1.79	

**Table 3 sensors-20-02128-t003:** Results of Code No. roughness measurement 178–601, serial no. 318411402, using contactless and contact methods.

Measurement Method	$R_{a}\;[µm]$	$R_{max}(\;R_{y})\;[µm]$
Reference data	2.94	9.3
Contact method (profilometer)	2.92	9.3
Contactless method (OCT)	2.87	10.2

**Table 4 sensors-20-02128-t004:** Results of roughness measurement obtained using a profilometer.

Parameter	Clean Substrate [µm]	SD	with Deposited Metallic Layer [µm]	SD
$S_{p}$	38.10	6.82	27.10	3.82
$S_{v}$	58.00	18.54	19.10	5.35
$S_{q}$	9.14	2.1	3.77	1.24
$S_{a}$	6.81	1.58	2.55	0.84

**Table 5 sensors-20-02128-t005:** Cordura area roughness parameters measured from OCT images.

Parameter	Clean Substrate [µm]	SD	with Deposited Metallic Layer [µm]	SD
$S_{p}$	77.15	30.37	46.08	10.49
$S_{v}$	59.30	11.08	44.40	10.16
$S_{q}$	24.57	6.17	16.96	2.23
$S_{a}$	20.02	4.95	13.66	1.72

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
