# Peer review of "Surface Morphology Analysis of Metallic Structures Formed on Flexible Textile Composite Substrates"

_sensors, 2020, doi:10.3390/s20072128_

Round 1

Reviewer 1 Report

This manuscript describes the usefulness of an optical technique called optical coherence tomography (OCT) to measure, in a noncontact manner, the surface roughness of metallized textiles.

The conceit of this manuscript is, frankly, a little hard to agree with and the projected impact on textile electronics is poorly described.

While I concede that it is cool to see OCT applied to metallized textile substrates, I cant see why confocal microscopy (another broadly-commercialized instrument) cant provide the same images and analysis as described in this paper. The main draw of OCT over confocal microscopy is the potential for OCT to actually peer into 3D structures (like cells, tissues,etc) and reveal an image of more than just the surface of the structure--no such clarification is seemingly provided here. If the images actually reveal areas where the metal coating deasheres from or is otherwise physically separated from the underlying textile substrate, I can see the power of this technique--these revelations are not present in this work, as far as I can see.

Moreover, the introduction and conclusions sections that discuss the motivation for this work repeatedly bring up the issue of micro-cracking,  and mechanical washing of metal coatings on textile, without actually showing how an OCT image can reveal this information. There is a lot of talk about metal nanostructures, etc, but I cant see how researchers can use an OCT image to predict whether a particular size range of nano/mesostructure will show more or less mechanical stability on underlying textile surfaces.

Author Response

Dear Reviewer,

We would like to thank you very much for your time you spent for correcting our paper and suggestions. They have helped us to improve the paper.

Below we have written answers for your comments. All of them are placed directly under the comments and they are highlighted in the text:

This manuscript describes the usefulness of an optical technique called optical coherence tomography (OCT) to measure, in a noncontact manner, the surface roughness of metallized textiles.

The conceit of this manuscript is, frankly, a little hard to agree with and the projected impact on textile electronics is poorly described.

While I concede that it is cool to see OCT applied to metallized textile substrates, I cant see why confocal microscopy (another broadly-commercialized instrument) cant provide the same images and analysis as described in this paper. The main draw of OCT over confocal microscopy is the potential for OCT to actually peer into 3D structures (like cells, tissues,etc) and reveal an image of more than just the surface of the structure--no such clarification is seemingly provided here. If the images actually reveal areas where the metal coating deasheres from or is otherwise physically separated from the underlying textile substrate, I can see the power of this technique--these revelations are not present in this work, as far as I can see.

Additional explanation of using OCT technique is added to the manuscript in the Introduction section:

“Roughness tests using a profilometer relate to a micrometer scale. Confocal microscopy measures unevenness in the sub-micrometer and nanometric scale with a resolution up to 0.1 µm/px. Therefore, confocal microscope measurement is not an alternative to profilometer measurement. Optical coherent tomography, due to its resolution, also measures roughness parameters in a micrometric scale. For this reason, it can be an alternative method to contact measurement using a profilometer. In addition, due to the flexibility of the measured substrate, the optical method using OCT has been select by the authors. The choice of the OCT method is made for economic reasons when no nanometer resolution of the profile is required. In this application confocal systems are usually very expensive. Real-time measurement is also possible for OCT due to flexible positioning of its scanning head over the surface of the material moving at a low speed [30].”

The mentioned reference has been added to the literature.

  1. Alarousu E., AlSaggaf A., Jabbour G. E.: Online monitoring of printed electronics by Spectral-Domain Optical Coherence Tomography; Scientific Reports, 3 (2013) 1562

Moreover, the introduction and conclusions sections that discuss the motivation for this work repeatedly bring up the issue of micro-cracking, and mechanical washing of metal coatings on textile, without actually showing how an OCT image can reveal this information. There is a lot of talk about metal nanostructures, etc, but I cant see how researchers can use an OCT image to predict whether a particular size range of nano/mesostructure will show more or less mechanical stability on underlying textile surfaces.

“Using the proposed optical method of assessing the surface profile, discontinuities and coming through defects of the metallic layer can be also observed, if they are at least the size of the device lateral resolution equal to 5.4 µm. They occur in locations where you can see lighter vertical lines of the echo created when IR beam penetrates under the material surface (Fig. 15b). The IR light cannot penetrate the continuous metal layer, so in the image in Fig. 15c the area below the surface border stays black. “

The above comment is an example of OCT using to detect defects in metallic structures (cracks and discontinuities in metallic structures). Such a comment was placed in the article below Fig. 15, in the paragraph 3.4.

All numbers of references have been corrected.

Attached please find the improved and corrected text of our article.

Reviewer 2 Report

In this manuscript the authors compare two different techniques for measuring textile roughness with and without a metal layer deposited on top. A non-contact technique is compared to a contact method.

The paper in general is well written, the state of the art and methods well-presented and the conclusions supported by the experimental data. I just have a few comments that would like to be answered before the manuscript is published:

  • Why the OCT technique was chosen over the other optical techniques? Does this technique have any advantage over other techniques or is it a completely new technique in the field that hasn’t been used in the past?
  • In line 375-377, the authors claim that Rmax value is overestimated because of “impurities and reflections from the shiny metallic surface”. Is this a general limitation of this technique? If a less reflective metal would be used, will this still be a limitation? I would appreciate 1-2 sentences discussing this.
  • In lines 417-439, the authors suggest that the OCT technique give larger roughness numbers because the profilometer is squeezing the polymeric textile fibers. This apparently doesn’t happen with the control measurement (paragraph 3.1). It is a bit confusing since I was thinking that the control experiments would be done with a similar material which is apparently not the case. It would be very informative if the authors could inform on the material used in the reference sample. Even better would be to compare mechanical parameters of both reference sample and Cordura such as the Young modulus. In this way, the authors could easily claim that Cordura is less rigid than the reference sample and demonstrate the validity of their argument. Anyway, not knowing the difference between the materials makes the conclusions not very well supported…
  • I have a general question about the usage of silver. Normally, silver doesn’t adhere well to other substrates. Typically, materials such as Cr and Ti are used as adhesion layers to increase the adhesion. From the methods, I see that the authors don’t use any adhesion layer. Is there any reason of not using an adhesion layer? Do the authors observe that the silver peels off the textile or start cracking as the sample is bent? Could the scalping of the silver be explained by poor adhesion of silver to the Cordura material?

There are also minor corrections that should be done:

  • Line 153, equation (3): I guess that the definition of Sa doesn’t have the square root.
  • Figure 3: the B box is misplaced as I am guessing from the inset that it should be on top of the sliding table D.
  • Line 329: I think the authors are referring to Figure 9a and not Figure 8a.
  • Figure 12 and 13: Could you please put the same color scale in the 4 figures? It is easier to compare the figures with each other if the same color scale is used.

Author Response

Dear Reviewer,

We would like to thank you very much for your time you spent for correcting our paper and suggestions. They have helped us to improve the paper.

Below we have written answers for your comments. All of them are placed directly under the comments and they are highlighted in the text:

In this manuscript the authors compare two different techniques for measuring textile roughness with and without a metal layer deposited on top. A non-contact technique is compared to a contact method.

The paper in general is well written, the state of the art and methods well-presented and the conclusions supported by the experimental data. I just have a few comments that would like to be answered before the manuscript is published:

  • Why the OCT technique was chosen over the other optical techniques? Does this technique have any advantage over other techniques or is it a completely new technique in the field that hasn’t been used in the past?

OCT can be an alternative method to contact measurement using a profilometer. In addition, due to the flexibility of the measured substrate, the optical method using OCT seems to be justified. Additionally, the OCT technique is economically justified in comparison to the confocal microscopy method ensuring the highest accuracy of profile analysis.

The Introduction section has been developed and the additional information about OCT has been written.

“Roughness tests using a profilometer relate to a micrometer scale. Confocal microscopy measures unevenness in the sub-micrometer and nanometric scale with a resolution up to 0.1 µm/px. Therefore, confocal microscope measurement is not an alternative to profilometer measurement. Optical coherent tomography, due to its resolution, also measures roughness parameters in a micrometric scale. For this reason, it can be an alternative method to contact measurement using a profilometer. In addition, due to the flexibility of the measured substrate, the optical method using OCT has been select by the authors. The choice of the OCT method is made for economic reasons when no nanometer resolution of the profile is required. In this application confocal systems are usually very expensive. Real-time measurement is also possible for OCT due to flexible positioning of its scanning head over the surface of the material moving at a low speed [30].”

Pros and cons of the OCT method compared to the profilometric method have been shown in Table 1 now.

The OCT method was used to test the quality of printed circuit boards online [30]. Comparative research was conducted by Czajkowski et al. using high resolution OCT [40]. Information was placed in the Introduction section – lines 117-122.

The mentioned references have been added to the literature:

  1. Alarousu E., AlSaggaf A., Jabbour G. E.: Online monitoring of printed electronics by Spectral-Domain Optical Coherence Tomography; Scientific Reports, 3 (2013) 1562
  2. Czajkowski J., Prykӓri T., Alarousu E., Palosaari J., Myllylӓ R.: Optical coherence tomography as a method of quality inspection for printed electronics products. Optical Review, 17 (2010) 3,: 257–262.

  • In line 375-377, the authors claim that Rmax value is overestimated because of “impurities and reflections from the shiny metallic surface”. Is this a general limitation of this technique? If a less reflective metal would be used, will this still be a limitation? I would appreciate 1-2 sentences discussing this.

Reflective surfaces can cause OCT image distortions over metal surface visible in the air layer in the form of bright vertical streaks. This phenomenon can be partially reduced by shifting the focus range towards the material surface. Then, however, the material border is slightly blurred. Individual, small reflections of IR light can appear over different surfaces and affect Rmax value. – such information is placed in the article in the section 3.1 Control measurement:

“Reflective surfaces can cause OCT image distortions over metal surface visible in the air layer in the form of bright vertical streaks. This phenomenon can be partially reduced by shifting the focus range towards the material surface. Then, however, the material border is slightly blurred. Individual, small reflections of IR light can appear over different surfaces and affect Rmax value.”

  • In lines 417-439, the authors suggest that the OCT technique give larger roughness numbers because the profilometer is squeezing the polymeric textile fibers. This apparently doesn’t happen with the control measurement (paragraph 3.1). It is a bit confusing since I was thinking that the control experiments would be done with a similar material which is apparently not the case. It would be very informative if the authors could inform on the material used in the reference sample. Even better would be to compare mechanical parameters of both reference sample and Cordura such as the Young modulus. In this way, the authors could easily claim that Cordura is less rigid than the reference sample and demonstrate the validity of their argument. Anyway, not knowing the difference between the materials makes the conclusions not very well supported…

The section: Control Measurement was only to calibrate both roughness determination methods which are presented in the article. To make the paper easier to read, the authors added information about the material from which the roughness reference sample was made.

The sentence “The reference sample is made of alloy stainless steel.” has been written in the 2.1. Section: Control measurement.

  • I have a general question about the usage of silver. Normally, silver doesn’t adhere well to other substrates. Typically, materials such as Cr and Ti are used as adhesion layers to increase the adhesion. From the methods, I see that the authors don’t use any adhesion layer. Is there any reason of not using an adhesion layer? Do the authors observe that the silver peels off the textile or start cracking as the sample is bent? Could the scalping of the silver be explained by poor adhesion of silver to the Cordura material?

The additional information has been added to the text:

“Thermal evaporation is one of the cheapest PVD technologies that takes place in a relatively short time. On the other hand, it is not possible to deposit materials that boil at high temperatures such as titanium (boiling point 3287°C) or chromium (2572°C). Therefore, the authors of the work did not use these materials to increase the adhesion of the silver coating to the substrate.”

The information about the boiling point for silver was also placed in the text in the section: Samples. – “(boiling point 2162°C), evaporated from the tungsten boat (melting point 3410°C)”

For better explanation the scope of the proposed optical method uses the following information is added to the text at the end of section 3.3:

“Using the proposed optical method of assessing the surface profile, discontinuities and coming through defects of the metallic layer can be also observed, if they are at least the size of the device lateral resolution equal to 5.4 µm. They occur in locations where you can see lighter vertical lines of the echo created when IR beam penetrates under the material surface (Fig. 15b). The IR light cannot penetrate the continuous metal layer, so in the image in Fig. 15c the area below the surface border stays black.”

In the conducted research, the separation of the silver layer from the substrate has not been observed. The contact profilometry method, due to the contact nature, caused mechanical peeling off of the metallic layer.

The following comment has been added to the Summary section: “The studies have been conducted on samples without cracks or delaminations to assess the profile of a uniform surface. Nevertheless, the measurement algorithm can take into account the location of defects in the case of their occurrence. Using this method, it is possible to assess the quality of the created layers by measuring the number and size of discontinuities, but only these extending from the metal surface to the textile substrate. This problem will be studied in the future.”

There are also minor corrections that should be done:

  • Line 153, equation (3): I guess that the definition of Sa doesn’t have the square root.

The square root symbol is a mistake created when rewriting the equation. As in equation (1) describing 1D case, there is no reason to insert a square root operation. The formula has been corrected.

  • Figure 3: the B box is misplaced as I am guessing from the inset that it should be on top of the sliding table D.

The figure has been corrected

  • Line 329: I think the authors are referring to Figure 9a and not Figure 8a.

The number of the Figure has been corrected

  • Figure 12 and 13: Could you please put the same color scale in the 4 figures? It is easier to compare the figures with each other if the same color scale is used.

Suggested Figures have been changed.

Figures 12 a and b have the same resolution and colour scale. If the same scale is used for the Figure 13 as for Figure 12, these images are illegible. For this reason, a different scale was used then, but the same for both Figures 13 a and b.

All numbers of references have been corrected.

Attached please find the improved and corrected text of our article.

Reviewer 3 Report

English Language was very good but some typo should be revised.

Summary is just a repetition of abstract. Please rewrite to explore/clarify the significance of the manuscript target.

Textile based sensor is a class of technical/smart textiles. Thus, authors should discuss briefly technical/smart textiles in introduction section and accordingly cite the following papers:

 Meyer, Jan, et al. "Design and modeling of a textile pressure sensor for sitting posture classification." IEEE Sensors Journal 10.8 (2010): 1391-1398. Ahmed, H., et al.  Mattmann, Corinne, Frank Clemens, and Gerhard Tröster. "Sensor for measuring strain in textile." Sensors 8.6 (2008): 3719-3732. Khattab, Tawfik A., et al. "Co-encapsulation of enzyme and tricyanofuran hydrazone into alginate microcapsules incorporated onto cotton fabric as a biosensor for colorimetric recognition of urea." Reactive and Functional Polymers 142 (2019): 199-206. Lee, Jaehong, et al. "Conductive fiber‐based ultrasensitive textile pressure sensor for wearable electronics." Advanced materials 27.15 (2015): 2433-2439.

Author Response

Dear Reviewer,

We would like to thank you very much for your time you spent for correcting our paper and suggestions. They have helped us to improve the paper.

Below we have written answers for your comments. All of them are placed directly under the comments and they are highlighted in the text:

English Language was very good but some typo should be revised.

Summary is just a repetition of abstract. Please rewrite to explore/clarify the significance of the manuscript target.

Abstract and Summary sections have been modified.

Among other additional information has been added to the summary section:

“The studies have been conducted on samples without cracks or delaminations to assess the profile of a uniform surface. Nevertheless, the measurement algorithm can take into account the location of defects in the case of their occurrence. Using this method, it is possible to assess the quality of the created layers by measuring the number and size of discontinuities, but only these extending from the metal surface to the textile substrate. This problem will be studied in the future.

The OCT method can be applied not only for conductive layers, but also for other surfaces, with particular emphasis on flexible materials. Due to the ability of the IR light beam to penetrate semi-translucent or turbid layers the OCT method can be applied to analyze the profiles of internal surfaces covered by thin polymer or textile layers. This feature enables the inline quality validation of textronic sensors in the industry production conditions.”

The sentence, showing the potential target, has been written in the abstract:

“Assessment of the surface profile of textile substrates and the thin films created on them, is important when estimating the contact angle, wetting behavior or mechanical durability of created metallic structure which can be used as the electrodes or elements of wearable electronics or textronics systems.”

Textile based sensor is a class of technical/smart textiles. Thus, authors should discuss briefly technical/smart textiles in introduction section and accordingly cite the following papers:

 Meyer, Jan, et al. "Design and modeling of a textile pressure sensor for sitting posture classification." IEEE Sensors Journal 10.8 (2010): 1391-1398. Ahmed, H., et al.  Mattmann, Corinne, Frank Clemens, and Gerhard Tröster. "Sensor for measuring strain in textile." Sensors8.6 (2008): 3719-3732. Khattab, Tawfik A., et al. "Co-encapsulation of enzyme and tricyanofuran hydrazone into alginate microcapsules incorporated onto cotton fabric as a biosensor for colorimetric recognition of urea." Reactive and Functional Polymers 142 (2019): 199-206. Lee, Jaehong, et al. "Conductive fiber‐based ultrasensitive textile pressure sensor for wearable electronics." Advanced materials 27.15 (2015): 2433-2439.

The authors have discussed briefly smart textiles used as the sensors in the introduction section.

The following information has been added to the paper:

“The sensor market is developing rapidly. According to Zion Market Research, the sensor market will be worth $ 11.2 billion in 2025 [11]. Many of the sensors are integrated with clothing and act as elements of wearable electronics systems, when they are built in clothes are always in the right place on the body. The way they are positioned ensures convenience and discretion for the user. Due to the small size and weight, wearing comfort is also ensured, and the user does not have to care about placing it. Textronic sensors must also have low power consumption, flexibility, as well reliable sensing performances. The sensors combined with clothes include the following types: gas [12], pressure [13], strain [14], temperature [15], sitting posture [16] or urea [17] detectors.”

The suggested references have been added to the reference list, positions 11-17.

All numbers of references have been corrected.

Attached please find the improved and corrected text of our article.

Reviewer 4 Report

Surface morphology analysis is one of the important subjects of material science and engineering. Ewa and coworkers reported an interesting works on the comparing methods for measuring selected morphological features on the surface of thin metallic layers. Their measurements were carried out at the micro scale using both optical coherent tomography (OCT) and the traditional contact method of using a profilometer. The experimental data is solid, the theoretical analysis seems resonable, and the whole discusions are good. It could be publised after minor considerations.

As the non-contact method they use in their work, I think they should focus on the OCT method, for examples, in the introduction part authors state some research status of OCT and its application in related fields.

In the introduction part, when comparing different test methods, it is better to use tables to show the data. It is easier for readers to see the advantages and disadvantages of different methods from the tables.

In 2.1 samples. what is the thickness of silver layer could it be estimated or measured in the process of preparation, and what is the effect of silver thickness on roughness? Can other conductive metals replace silver? Can it be non-metallic oxide such as ITO? Are other oxides OK? Whether organic surface can be measured directly?

Author Response

Dear Reviewer,

We would like to thank you very much for your time you spent for correcting our paper and suggestions. They have helped us to improve the paper.

Below we have written answers for your comments. All of them are placed directly under the comments and they are highlighted in the text:

Surface morphology analysis is one of the important subjects of material science and engineering. Ewa and coworkers reported an interesting works on the comparing methods for measuring selected morphological features on the surface of thin metallic layers. Their measurements were carried out at the micro scale using both optical coherent tomography (OCT) and the traditional contact method of using a profilometer. The experimental data is solid, the theoretical analysis seems resonable, and the whole discusions are good. It could be publised after minor considerations.

As the non-contact method they use in their work, I think they should focus on the OCT method, for examples, in the introduction part authors state some research status of OCT and its application in related fields.

According to the reviewer’s suggestion the additional information has been placed in the Introduction section:

“OCT is a high-resolution imaging technique, which provides depth profiles of inhomogeneous and turbid materials in a contactless and non-destructive manner [31]. Generation of OCT images is based on the measure of the magnitude and the time delay of light reflected back from an investigated sample via an interferometric approach. Within the sample the light is back reflected at scattering particles and interfaces between materials with different refractive indices. Through the interferometric detection, OCT allows for a measurement of optical pathlengths, which are related to the geometric structure of the sample. It offers non-invasive optical imaging of structures located in a sample with excellent spatial resolution (<10 mm).

OCT is suitable for testing both biological structures and artificial materials, but in the initial period of development of this method, the focus was almost exclusively on organic systems research.

For over 15 years the OCT is increasingly being used for the study of structures of materials. It tenders an approach to quantify both surface and internal (overall and local) properties of paper [32], silicon integrated-circuits [33], fiber composites [34], dental composites [35], various kinds of coating, e.g. in pharmaceuticals [36], and electrical components [37].

OCT is used not only to study deep structures but due to the ability to accurately three-dimensional mapping the topology of samples can be used to visual quality inspection of surface layer or study surface phenomena such as wettability of materials [38,39].

The first study demonstrating the application of OCT in printed functional materials and printed electronics was published by Czajkowski et al. in 2010 [40]. The authors applied ultra-high resolution time domain OCT (UHR TD-OCT) to evaluate the internal structure of epoxy embedded RF-antenna. The same group published later the use of UHR TD-OCT to study quality of protective films used in printed electronics [41]. Thrane et al. in 2012 [42] demonstrated the use of TD-OCT in imaging the multilayer structure and identifying defects of R2R coatings in a polymer solar cells.

Modern OCT tools offer a speed enabling online monitoring of printed devices. Alarousu et. al. 2013 [30] used spectral domain OCT (SD-OCT) to monitor structural surface properties of a moving sample of silver-based electrodes printed on a flexible PET plastic substrate. This reveals the advantage of non-invasive OCT inspection over traditional surface testing methods, especially contact profilometers, which have limitations in the respect of real time working.”

In the introduction part, when comparing different test methods, it is better to use tables to show the data. It is easier for readers to see the advantages and disadvantages of different methods from the tables.

According to the reviewer’s suggestion the disadvantages and advantages of presented method are presented in the following table:

Table 1. Comparison of contact method and optical coherence tomography (OCT) method

Advantages

Disadvantages

Contact method

·   full reflection of the measured surface

·   no influence of surface optical characteristics

·   good penetration of the tested surface

·   high sampling resolution up to 0.1 µm/px

·        the destructive impact of the measuring needle on the tested surface

·        the needle leaves traces in the materials of low hardness

·        high level of noise generated by the mechanical system

·        long measurement time, movement speed of the needle as low as 0.1 mm/s

·        possible physicochemical reactions may occur between the needle and the tested material

OCT optical method

·   no impact of the measurement on the tested surface

·       high influence of surface reflectance characteristic on the measurement results

·   short measurement time with scanning speed »10 mm/s

·       lower sampling resolution of 5.4 µm/px in the case of Spark OCT scanning

·   no traces left on low hardness surfaces

·   no reaction physical or chemical reaction between low power laser needle (2mW) and the tested surface

·       high level of speckle noise in typical OCT images

·       possible false peaks and gaps in the surface profile requiring approximation

·   easy detection of breaks and holes in the metal coating of a textile by the IR light penetrating below the surface

·   detection of internal surfaces under semi-translucent coatings

In 2.1 samples. what is the thickness of silver layer could it be estimated or measured in the process of preparation, and what is the effect of silver thickness on roughness? Can other conductive metals replace silver? Can it be non-metallic oxide such as ITO? Are other oxides OK? Whether organic surface can be measured directly

Additionally, the description of the OCT has been expanded in the Introduction section:

“OCT is a high-resolution imaging technique, which provides depth profiles of inhomogeneous and turbid materials in a contactless and non-destructive manner [31]. Generation of OCT images is based on the measure of the magnitude and the time delay of light reflected back from an investigated sample via an interferometric approach. Within the sample the light is back reflected at scattering particles and interfaces between materials with different refractive indices. Through the interferometric detection, OCT allows for a measurement of optical pathlengths, which are related to the geometric structure of the sample. It offers non-invasive optical imaging of structures located in a sample with excellent spatial resolution (<10 mm).

OCT is suitable for testing both biological structures and artificial materials, but in the initial period of development of this method, the focus was almost exclusively on organic systems research.

For over 15 years the OCT is increasingly being used for the study of structures of materials. It tenders an approach to quantify both surface and internal (overall and local) properties of paper [32], silicon integrated-circuits [33], fiber composites [34], dental composites [35], various kinds of coating, e.g. in pharmaceuticals [36], and electrical components [37].

OCT is used not only to study deep structures but due to the ability to accurately three-dimensional mapping the topology of samples can be used to visual quality inspection of surface layer or study surface phenomena such as wettability of materials [38,39].

The first study demonstrating the application of OCT in printed functional materials and printed electronics was published by Czajkowski et al. in 2010 [40]. The authors applied ultra-high resolution time domain OCT (UHR TD-OCT) to evaluate the internal structure of epoxy embedded RF-antenna. The same group published later the use of UHR TD-OCT to study quality of protective films used in printed electronics [41]. Thrane et al. in 2012 [42] demonstrated the use of TD-OCT in imaging the multilayer structure and identifying defects of R2R coatings in a polymer solar cells.

Modern OCT tools offer a speed enabling online monitoring of printed devices. Alarousu et. al. 2013 [30] used spectral domain OCT (SD-OCT) to monitor structural surface properties of a moving sample of silver-based electrodes printed on a flexible PET plastic substrate. This reveals the advantage of non-invasive OCT inspection over traditional surface testing methods, especially contact profilometers, which have limitations in the respect of real time working. “

The following text has been placed in the paper in the section 2.1 Samples:

“The thickness of the deposited metallic layer was measured indirectly. In the vacuum deposition process, the laboratory glass was placed next to the tested substrate in the same technology process. The metal was deposited on both surfaces. Then, using the contact profilometer, the thickness of the layer on the glass (treated as the reference one) was assessed as the height of the measuring needle step. The thickness of the silver applied layer was estimated as 250 nm.”

And in the Summary:

“The OCT method can be applied not only for conductive layers, but also for other surfaces, with particular emphasis on flexible materials. Due to the ability of the IR light beam to penetrate semi-translucent or turbid layers the OCT method can be applied to analyze the profiles of internal surfaces covered by thin polymer or textile layers. This feature enables the inline quality validation of textronic sensors in the industry production conditions “

The mentioned references have been added to the literature:

  1. Alarousu E., AlSaggaf A., Jabbour G. E.: Online monitoring of printed electronics by Spectral-Domain Optical Coherence Tomography; Scientific Reports, 3 (2013) 1562
  2. Fercher A. F.: Optical coherence tomography – development, principles, applications: Med. Phys. 20, (2010): 251–276.
  3. Prykäri T., Czajkowski J., Alarousu E., Myllylä R.: Optical coherence tomography as an accurate inspection and quality evaluation technique in paper industry: Optical Review (2010), 17: 218–222.
  4. Serrels K.A., Renner M.K., Reid D.T.: Optical coherence tomography for non- destructive investigation of silicon integrated-circuits. Eng. (2010), 87: 1785–1791.
  5. Stifter D.: Beyond biomedicine: a review of alternative applications and developments for optical coherence tomography; Phys. B. (2007), 8:,337–357.
  6. Braz A. K., Kyotoku B. B., Braz R., Gomes A. S.: Evaluation of crack propagation in dental composites by optical coherence tomography. Dental Materials, 25(1) (2009): 74–79.
  7. Markl D., Zettl M., Hannesschläger G., Sacher S., Leitner M., Buchsbaum A., Khinast J. G. Calibration-freein-line monitoring of pellet coating processes via optical coherence tomography. Chemical Engineering Science, 125, (2015): 200–208.
  8. Cho N. H., Jung U., Kim S., Kim J.: Non-Destructive Inspection Methods for LEDs Using Real-Time Displaying Optical Coherence Tomography; Sensors-Basel 12, (2012): 10395–10406
  9. Fabritius T., Myllylä R., Makita S., Yasuno, Y.: Wettability characterization method based on optical coherence tomography imaging; Optics Express 18 (2010) 22859. doi:10.1364/oe.18.022859
  10. Gocławski J., Sekulska-Nalewajko J., Strzelecki B., Romanowska I. Wettability analysis method for assessing the effect of chemical pretreatment on brown coal biosolubilization by Gordonia alkanivorans Fuel, 256 (2019) 115927, https://doi.org/10.1016/j.fuel.2019.115927
  11. Czajkowski J., Prykӓri T., Alarousu E., Palosaari J., Myllylӓ: Optical coherence tomography as a method of quality inspection for printed electronics products. Optical Review, 17, 3 (2010): 257–262.
  12. Czajkowski, J. et al. Ultra-high resolution optical coherence tomography for encapsulation quality inspection. Appl Phys B-Lasers O 105 (2011) 29, 649–657
  13. Thrane L., Jorgensen T. M., Jorgensen M., Krebs F. C.: Application of optical coherence tomography (OCT) as a 3-dimensional imaging technique for roll-to-roll coated polymer solar cells. Sol Energ Mat Sol C 97 (2012)181–185

All numbers of references have been corrected.

Attached please find the improved and corrected text of our article.

Round 2

Reviewer 1 Report

I think the authors addressed most comments.